# Z-SASLM: Zero-Shot Style-Aligned SLI Blending Latent Manipulation

## Abstract

*We introduce Z-SASLM, a Zero-Shot Style-Aligned SLI (Spherical Linear Interpolation) Blending Latent Manipulation pipeline that overcomes the limitations of current multi-style blending methods. Conventional approaches rely on linear blending, assuming a flat latent space leading to suboptimal results when integrating multiple reference styles. In contrast, our framework leverages the non-linear geometry of the latent space by using SLI Blending to combine weighted style representations. By interpolating along the geodesic on the hypersphere, Z-SASLM preserves the intrinsic structure of the latent space, ensuring high-fidelity and coherent blending of diverse styles—all without the need for fine-tuning. We further propose a new metric, Weighted Multi-Style DINO VIT-B/8, designed to quantitatively evaluate the consistency of the blended styles. While our primary focus is on the theoretical and practical advantages of SLI Blending for style manipulation, we also demonstrate its effectiveness in a multi-modal content fusion setting through comprehensive experimental studies. Experimental results show that Z-SASLM achieves enhanced and robust style alignment. The code will be made publicly available upon completion of the review process.*

## 1. Introduction

Text-to-image generation has advanced rapidly, moving from GANs[1] and its numerous variants[2–7] to powerful diffusion-based models [8, 9]. The rapid development of large-scale text-to-image (T2I) models such as DALL·E [10], MidJourney [11], Stable Diffusion [12] and others [13–15] has revolutionized creative industries by enabling the generation of high-quality, diverse visual outputs from textual descriptions. Despite these advances, achieving consistent *Style Alignment*—the ability to maintain a coherent visual style across multiple generated images—remains a significant challenge. For example, a designer aiming to blend the distinct aesthetics of cubism and baroque across visuals must often resort to fine-tuning models on specific styles. This approach limits the blending of the styles

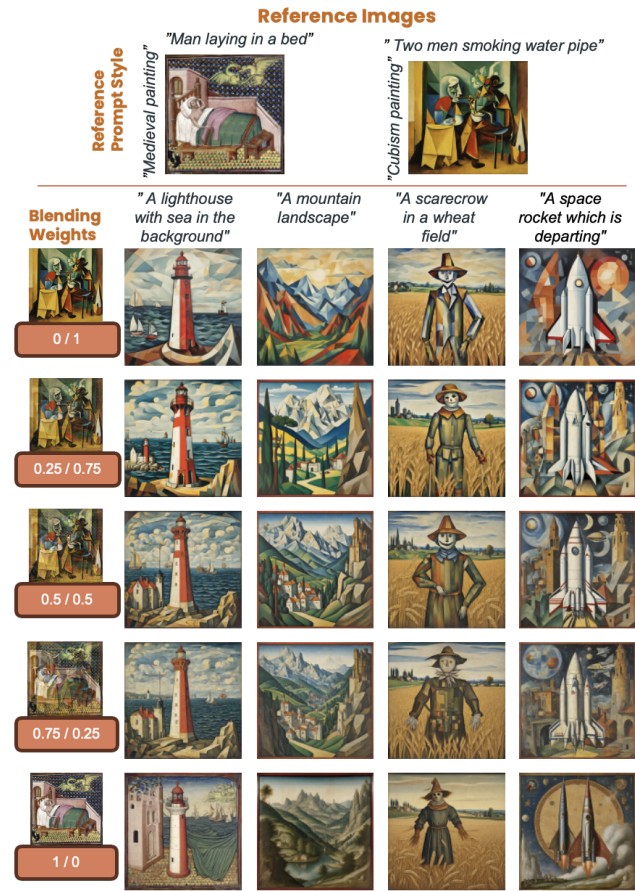

Figure 1. Medieval-Cubism SLI Blending (2-styles)

present in the fine-tuning dataset and confines current methods to single-style references.

In this work, we propose a novel *Z-SASLM* architecture based on *Spherical Linear Interpolation (SLI) Blending* approach for multi-reference style conditioning, a technique able to interpolate along the geodesic on the hypersphere, preserving the image manifold's geometric properties and ensuring smooth, coherent blending between styles. Importantly, our method eliminates the need for fine-tuning, enabling zero-shot style alignment directly during generation.

**Paper Contributions.** Our main contributions are the following:

- Z-SASLM architecure based on SLI Blending for Multi-Reference Style Conditioning: We introduce a SLI-based blending technique that leverages the latent space's non-linear geometry to combine multiple reference-style images in a weighted manner. This approach overcomes the limitations of linear blending and bypasses the need for fine-tuning.

- Weighted Multi-Style DINO VIT-B/8 Weighted Metric: We propose a new metric designed to evaluate the consistency of the style in a set of generated images, effectively quantifying the contributions of multiple blended styles.

- Multi-Modal Content Fusion Ablation: We conduct comprehensive ablation studies using multi-modal content fusion—integrating image, audio, and other modalities—to validate the improvements in style alignment even in a multi-modal setting.

## 2. Related Work

**Style Alignment in Text-to-Image Models.** Despite significant advances by models such as DALL·E [10], Stable Diffusion [12], and Imagen [8], achieving consistent style alignment remains a challenging task. Most state-of-the-art T2I models are designed for single-image generation and optimized for a single style reference, often resulting in inconsistencies when generating a series of images. Early approaches to style alignment can be broadly divided into *Fine-Tuning-Based* and *Latent Space Manipulation* methods. Fine-tuning techniques, like StyleDrop [16] and DreamBooth [17], require adapting the model to specific styles, which is computationally intensive, restricts the range of achievable styles, and limits scalability. Latent space manipulation methods, exemplified by StyleGAN [18] and Style-FiT [19], enable zero-shot style transfer but are predominantly designed for single style adaptation and often struggle with content-style disentanglement when blending multiple styles. In essence, while these methods generate individual images conditioned on a style reference, they do not enforce uniform style consistency across a set of generated images—as if they were all created by the same artist. Approaches such as StyleAligned[20] is the most similar approach to our idea, using shared attention mechanisms to enforce style consistency in a zero-shot manner; however, they remain confined to a single style reference and do not support weighted blending of multiple styles.

**Multi-Reference Style Blending.** Traditional methods to blend multiple reference styles, such as those employed in StyleGAN2-ADA [21], permit style mixing by directly manipulating the latent space, but they primarily rely on simple linear combinations and are designed for GAN-based architectures. These approaches suffer from several limitations: they assume a flat (Euclidean) latent space that fails to capture the curved geometry of high-dimensional representations, often resulting in abrupt transitions, artifacts, and incoherent style mixtures. In contrast, our work introduces a multireference-weighted style blend framework tailored to diffusion-based models, with at its core, SLI Blending, which interpolates along the geodesic on the hypersphere. This approach preserves the intrinsic structure of the latent manifold, ensuring that the weighted combination of style representations remains within a semantically meaningful region. By overcoming the inherent limitations of conventional, linear-based methods, our SLI Blending technique achieves a high-fidelity, coherent fusion of diverse styles, enabling smooth and consistent multi-style integration.

**Multi-Style Consistency Evaluation.** Evaluating the consistency of blended styles poses unique challenges. While metrics such as DINO ViT [22] have been successfully employed to assess style consistency, they are typically tailored to single-style scenarios and do not account for the nuances of weighted multi-style blending. To this end, we propose the Weighted Multi-Style DINO VIT-B/8 metric, an enhanced evaluation tool that extends traditional DINO ViT metrics to effectively quantify the coherence and contribution of multiple blended styles.

## 3. Method Overview

The Z-SASLM architecture, illustrated in Fig. 2, consists of three primary modules—Reference Images Encoding & Blending, Text Encoding, and the StyleAligned Image Generation process—augmented by an optional Multi-Modal Content Fusion module. When this optional module is not employed, a simple caption (denoted as *Single-Content Textual prompt*) indicating the desired scene we want to generate can serve as a substitute. The Multi-Modal Content Fusion module, on the other hand, provides a *Multi-Content Textual Prompt* that can aggregate diverse inputs such as images, audio, music, and weather data. It utilizes T5-based rephrasing to merge these modalities into a unified textual prompt, which is then forwarded to the Text Encoding module. Here, the prompt (either coming from the Multi-Modal Content Fusion module or from a simple caption) is tokenized and encoded using CLIP, producing embeddings for both positive and negative prompts. In parallel, the Reference Image Encoding & Blending module extracts latent vectors from multiple reference styles via a VAE encoder. These latent vectors are blended using our proposed SLI (Spherical Linear Interpolation) Blending approach. Finally, the unified style representation is combined with the textual embeddings and provided to the StyleAligned image generation process, resulting in images that exhibit coherent and consistent style alignment.

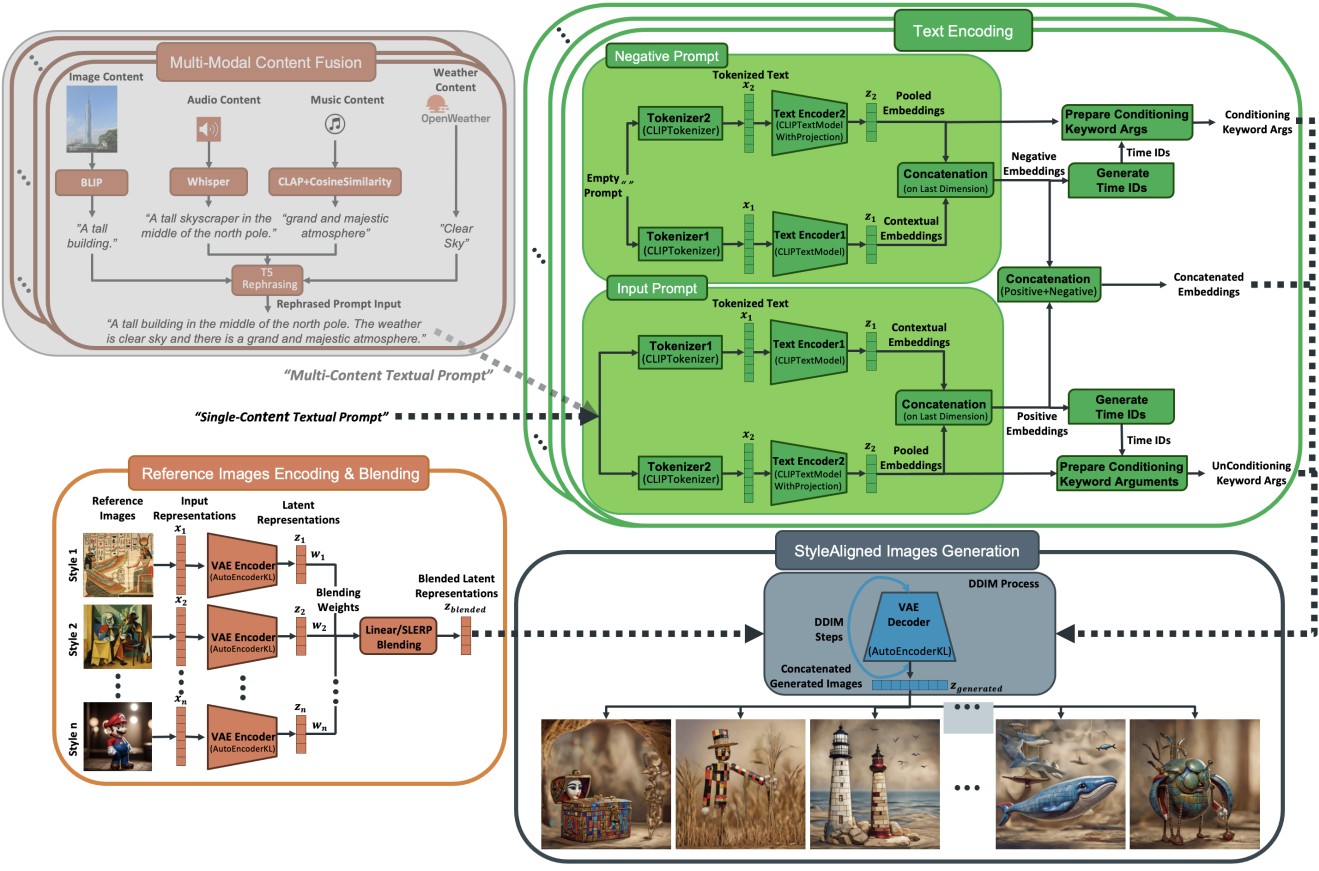

Figure 2. Overview of the Z-SAMB Architecture.

## 3.1. Background

To achieve style alignment during generation without fine-tuning, we adopt key components from StyleAligned [20]: Adaptive Instance Normalization (AdaIN) and Shared Attention.

**Adaptive Instance Normalization (AdaIN).** AdaIN [23] aligns the feature statistics of a generated image with those of a reference style image. Let $g = f(\mathbf{g})$ and $s = f(\mathbf{s})$ be the feature maps for the generated image and reference style, respectively, and define $\mu(\cdot)$ and $\sigma(\cdot)$ as the mean and standard deviation. Then AdaIN is:

$$\text{AdaIN}(g, s) = \sigma(s)\left(\frac{g - \mu(g)}{\sigma(g)}\right) + \mu(s) \quad (1)$$

**Shared Attention.** Shared Attention propagates style information across multiple images by computing pairwise similarities. For a set $\{\mathbf{x}_1, \ldots, \mathbf{x}_n\}$, with query $Q(\cdot)$, key $K(\cdot)$, and value $V(\cdot)$ features, the attention weights are:

$$A_{ij} = \frac{\exp\left(Q(f(\mathbf{x}_i)) \cdot K(f(\mathbf{x}_j))^\top / \sqrt{d}\right)}{\sum_{k=1}^{n} \exp\left(Q(f(\mathbf{x}_i)) \cdot K(f(\mathbf{x}_k))^\top / \sqrt{d}\right)} \quad (2)$$

which update the style representation as:

$$f(\mathbf{x}_i)' = \sum_{j=1}^{n} A_{ij} V\left(f(\mathbf{x}_j)\right) \quad (3)$$

In summary, queries and keys are first normalized via AdaIN using the reference style's statistics:

$$\hat{Q}_i = \text{AdaIN}\left(Q_i, Q_{\text{ref}}\right), \quad \hat{K}_i = \text{AdaIN}\left(K_i, K_{\text{ref}}\right) \quad (4)$$

The final style-aligned attention map is then computed by concatenating keys and values from both the reference image and the current image:

$$\text{Attention}\left(\hat{Q}_i, [K_{\text{ref}}, \hat{K}_i], [V_{\text{ref}}, V_i]\right) \quad (5)$$

By incorporating these mechanisms into our architecture, we enable zero-shot style alignment during generation, eliminating the need for fine-tuning.

## 3.2. Multi-Reference Weighted Style Blending

Inspired by the style-mixing approach of StyleGAN2-ADA [21], which linearly interpolates latent codes for GAN-based models, we adapt a similar idea to diffusion-based

models. However, because diffusion models typically operate in a non-Euclidean latent space, naively applying linear interpolation can lead to suboptimal or inconsistent style transitions. In this section, we first present linear blending as a baseline, then introduce our Spherical Linear Interpolation (SLI) Blending to address the limitations of linear mixing.

### 3.2.1. Linear Weighted Style Blending

Suppose we have $k$ reference style images $\{\mathbf{s}_1, \mathbf{s}_2, \ldots, \mathbf{s}_k\}$ mapped to latent vectors $\{\mathbf{z}_1, \mathbf{z}_2, \ldots, \mathbf{z}_k\}$, each weighted by $\{w_1, w_2, \ldots, w_k\}$ with $\sum_{i=1}^{k} w_i = 1$ A straightforward approach, similar to StyleGAN2-ADA [21], is to form a *linear* combination of these style vectors:

$$\mathbf{z}_{\text{blend}}^{\text{linear}} = \sum_{i=1}^{k} w_i \, \mathbf{z}_i \qquad (6)$$

We then use $\mathbf{z}_{\text{blend}}$ as a conditioning latent in the diffusion process. Although simple and intuitive, linear interpolation assumes a flat (Euclidean) geometry; in the curved, high-dimensional latent spaces of diffusion models, it can yield abrupt transitions or washed-out stylistic details. Consequently, by making several experiments, we often observe noticeable artifacts in the generated images, as the number of reference images augments, as illustrated in Fig. 4. These artifacts arise because the linear path between style vectors frequently falls off the meaningful regions of the latent manifold, causing a loss of fine-grained stylistic cues and reducing overall fidelity.

### 3.2.2. SLI Blending

We propose using *Spherical Linear Interpolation (SLI) Blending* to better preserve style fidelity. The idea of spherical interpolation was first introduced by Shoemake *et al.* [24] and consists in following the geodesic on a hypersphere rather than a straight line in Euclidean space, as illustrated in Fig. 3. By respecting the manifold's curvature, SLI produces smoother and more coherent blends of multiple styles compared to linear interpolation. Linear blending computes the weighted sum which, even if $\mathbf{z}_1$ and $\mathbf{z}_2$ are unit vectors, does not guarantee that $\mathbf{z}_{\text{blend}}^{\text{linear}}$ will also be of unit length. This deviation from the hypersphere can push the result away from high-density regions of the latent space, resulting in abrupt transitions and loss of fine stylistic details. Mathematically, for two unit vectors with angular separation $\omega$, their Euclidean (chord) distance is:

$$d_{\text{linear}} = 2\sin\left(\frac{\omega}{2}\right) \qquad (7)$$

while the true geodesic (arc) distance on the hypersphere is:

$$d_{\text{geodesic}} = d_{\text{sli}} = \omega \qquad (8)$$

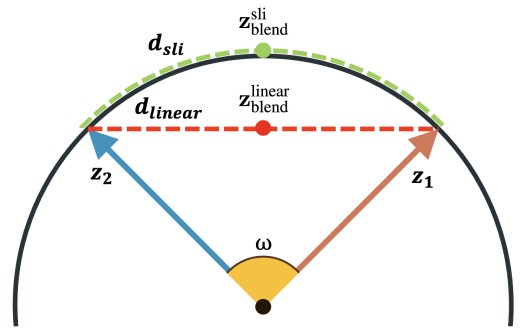

Figure 3. 2-style Linear vs. SLI in the latent space Illustration. The chord (in red) represents naive linear blending between two reference vectors $\mathbf{z}_1$ and $\mathbf{z}_2$, yielding $\mathbf{z}_{\text{blend}}^{\text{lin}}$. In contrast, Spherical Linear Interpolation (in green) follows the geodesic on the unit sphere, producing $\mathbf{z}_{\text{blend}}^{\text{SLI}}$. The angle $\omega$ denotes the separation between $\mathbf{z}_1$ and $\mathbf{z}_2$, with $d_{\text{linear}} < d_{\text{sli}}$ on the unit circle.

Since $2\sin\left(\frac{\omega}{2}\right) < \omega$ for $\omega > 0$, linear interpolation underestimates the true separation between styles. In contrast, SLI precisely follows the arc, ensuring that the interpolation accurately reflects the perceptual distance between the styles.

**Definition.** Starting with the simplest case in which we have *two style vectors* $\mathbf{z}_1$ and $\mathbf{z}_2$ with weights $w_1$ and $w_2$, let

$$t = \frac{w_2}{w_1 + w_2}, \quad \omega = \arccos\left(\frac{\mathbf{z}_1 \cdot \mathbf{z}_2}{\|\mathbf{z}_1\| \, \|\mathbf{z}_2\|}\right) \qquad (9)$$

Assuming that the latent vectors are normalized ($\|\mathbf{z}_1\| = \|\mathbf{z}_2\| = 1$), SLI is defined as:

$$\text{SLI}(t, \mathbf{z}_1, \mathbf{z}_2) = \frac{\sin\big((1-t)\,\omega\big)}{\sin(\omega)} \, \mathbf{z}_1 + \frac{\sin\big(t\,\omega\big)}{\sin(\omega)} \, \mathbf{z}_2 \qquad (10)$$

When $w_1 = w_2$, $t = 0.5$ yields an equal contribution; otherwise, the interpolation skews toward the style with the higher weight.

For $k > 2$ styles, we can simply extend SLI blending by iteratively combining style vectors. However, because SLI is a nonlinear operation defined on the hypersphere, its iterative application is not associative; that is, the final blended vector can depend on the order in which the style vectors are combined. To address this, we adopt a semantically meaningful ordering strategy by *sorting* the *reference styles* in *descending order* of their *weights*. This way, the most dominant styles are incorporated first, ensuring that their influence is robustly preserved in the cumulative blend, providing a stable, reproducible blending sequence and mitigating potential artifacts due to arbitrary ordering. More formally, let $\mathbf{z}_1, \mathbf{z}_2, \ldots, \mathbf{z}_k$ be the latent style vectors sorted such that $w_1 \geq w_2 \geq \cdots \geq w_k$. If $\mathbf{z}_{1\ldots i}$ denotes the cumulative

Table 1. DDIMScheduler Configuration

| Parameter | Value |
|---|---|
| beta_start | 0.00085 |
| beta_end | 0.012 |
| beta_schedule | scaled_linear |
| clip_sample | False |
| set_alpha_to_one | False |

Table 2. SDXL Pipeline Configuration

| Parameter | Value |
|---|---|
| torch_dtype | torch.float16 |
| variant | fp16 |
| use_safetensors | True |
| scheduler | DDIMScheduler |
| device | cuda |
| num_inference_steps | 50 |

blend of the first $i$ styles, we incorporate the $(i+1)$-th style with weight $w_{i+1}$ as follows:

$$\mathbf{z}_{1\cdots(i+1)} \;=\; \mathrm{SLI}\Big(\frac{w_{i+1}}{\sum_{m=1}^{i+1} w_m},\; \mathbf{z}_{1\cdots i},\; \mathbf{z}_{i+1}\Big) \qquad (11)$$

Repeating this process yields the final blended latent vector $\mathbf{z}_{\mathrm{blend}}^{\mathrm{sli}}$.

### 3.2.3. Dynamic Style-Aligned Arguments Scaling

Even with SLI, certain styles can dominate the blending process due to their more pronounced activations in the latent space. We refer to styles that tend to generate such strong responses as "famous," while those with standard activations are considered "normal." In our approach, we discriminate between these two categories by examining the norm of the latent key $\mathbf{k}$ for each style. Specifically, if $\|\mathbf{k}\|$ exceeds a predefined threshold $T = 0.5$, the style is classified as famous. This criterion is well-justified because the dot product in the attention mechanism scales with the norms:

$$\langle Q, K \rangle \propto \|Q\| \, \|K\| \qquad (12)$$

so a higher $\|\mathbf{k}\|$ directly results in disproportionately high attention scores, causing famous styles to overshadow others. This issue was also observed in the StyleAligned paper [20]. To mitigate this imbalance, we rescale the attention scores for each style reference by applying a style-dependent normalization. Specifically, we adjust the original attention score $A_{\mathrm{original}}$ as:

$$A_{\mathrm{normalized}} = A_{\mathrm{original}} \times \sigma + \mu \qquad (13)$$

where the shift $\mu$ and scale $\sigma$ are determined by the style's classification:

$$\{\mu, \sigma\} = \begin{cases} \{\log(2),\, 1\} & \text{style == normal ("n")} \\ \{\log(1),\, 0.5\} & \text{style == famous ("f")} \end{cases} \qquad (14)$$

This normalization dampens the excessive influence of famous styles by reducing their scale while preserving or even slightly boosting the contribution of normal styles through an appropriate shift.

### 3.3. Weighted Multi-Style DINO VIT-B/8

Evaluating style consistency in text-to-image (T2I) models requires a robust metric that captures both fine-grained visual details and higher-level semantic style characteristics.

The DINO (Distillation with No Labels) VIT-B/8 model [22]—a self-supervised vision transformer—has proven effective in this regard. Thanks to its multi-headed self-attention layers and an 8-patch input scheme, DINO VIT-B/8 extracts feature representations that encapsulate subtle stylistic nuances without requiring manual labels.

To assess the alignment between a generated image and a reference style, we first compute the cosine similarity between their corresponding feature embeddings. Let $\mathbf{z}_{\mathrm{gen}}$ denote the embedding of the generated image and $\mathbf{z}_{\mathrm{ref}}$ that of a reference style image. The cosine similarity is defined as:

$$\mathrm{CS}(\mathbf{z}_{\mathrm{gen}}, \mathbf{z}_{\mathrm{ref}}) = \frac{\mathbf{z}_{\mathrm{gen}} \cdot \mathbf{z}_{\mathrm{ref}}}{\|\mathbf{z}_{\mathrm{gen}}\| \, \|\mathbf{z}_{\mathrm{ref}}\|} \qquad (15)$$

While DINO VIT-B/8 excels at comparing a generated image against a single style reference, it does not directly support the evaluation of multi-style consistency. To address this, we extend the conventional cosine similarity into a weighted multi-style metric. Given the feature embeddings of $k$ reference style images and a generated image, we define the weighted multi-style similarity score as:

$$S_{\mathrm{multi\text{-}style}}(\mathbf{z}_{\mathrm{gen}}, \mathbf{z}_1, \ldots, \mathbf{z}_k) = \sum_{i=1}^{k} w_i \cdot \mathrm{CS}(\mathbf{z}_{\mathrm{gen}}, \mathbf{z}_i) \qquad (16)$$

This formulation effectively combines the individual style alignments into a single metric that reflects the overall style consistency of the generated image with respect to multiple references. More specifically, if we denote the cosine similarity for each reference as $s_i = \mathrm{CS}(\mathbf{z}_{\mathrm{gen}}, \mathbf{z}_i)$, then the final metric can be expressed as a weighted average:

$$S_{\mathrm{multi\text{-}style}} = \sum_{i=1}^{k} w_i \, s_i \qquad (17)$$

## 4. Experiments

We conduct our experiments on Stable Diffusion XL (SDXL), specifically the pre-trained model *'stabilityai/stable-diffusion-xl-base-1.0'* [25], following the setup of Hertz *et al*. [20]. However, our approach is model-agnostic and can be adapted to other diffusion-based or generative models. We use a DDIM (Denoising

| Style Weights | Linear (StyleGAN2-ADA[21]-adapted) | | | | Z-SASLM (Ours) | | | |
|---|---|---|---|---|---|---|---|---|
| $\{w_{med}, w_{cub}\}$ | $MS_{med}$ | $MS_{cub}$ | $WMS_{DINO\text{-}ViT\text{-}B/8}$ | $CLIP_{score}$ | $MS_{med}$ | $MS_{cub}$ | $WMS_{DINO\text{-}ViT\text{-}B/8}$ | $CLIP_{score}$ |
| $\{0, 1\}^*$ | - | 0.47552 | 0.47552 | 0.30280 | - | 0.47552 | 0.47552 | 0.30280 |
| $\{0.15, 0.85\}$ | 0.32466 | 0.42683 | 0.41151 | **0.31534** | 0.32595 | 0.47072 | **0.44900** | 0.31049 |
| $\{0.25, 0.75\}$ | 0.35550 | 0.42250 | 0.40575 | 0.31420 | 0.33046 | 0.45447 | **0.42347** | **0.31657** |
| $\{0.5, 0.5\}$ | 0.34905 | 0.37881 | 0.36393 | 0.29232 | 0.36150 | 0.42156 | **0.39153** | **0.31434** |
| $\{0.75, 0.25\}$ | 0.35798 | 0.38327 | **0.36430** | 0.31752 | 0.34648 | 0.35099 | 0.34760 | **0.31911** |
| $\{0.85, 0.15\}$ | 0.35513 | 0.40860 | 0.36315 | **0.32381** | 0.36513 | 0.38286 | **0.36779** | 0.31499 |
| $\{1, 0\}^*$ | 0.29891 | - | 0.29891 | 0.30570 | 0.29891 | - | 0.29891 | 0.30570 |

Table 3. Weighted Multi-Style DINO VIT-B/8: Linear vs. Z-SASLM. *: Rows $\{0, 1\}$ and $\{1, 0\}$ indicate no blending (i.e., StyleAligned [20]). We omit from the comparison the basic SDXL results as we are comparing only style-alignment results.

Diffusion Implicit Models) [26] scheduler (configuration in Tab. 1) and initialize the SDXL pipeline with parameters in Tab. 2. All experiments are performed on GPUs in mixed-precision mode for efficiency.

## 4.1. Weighted Multi-Style DINO VIT-B/8: Linear vs SLI Interpolation

We first compare our SLI Blending against the simpler Linear Blending approach, which is inspired by the linear interpolation used in StyleGAN2-ADA [21] but extended to diffusion models. To evaluate style consistency, we use the proposed Weighted Multi-Style DINO VIT-B/8 metric (indicated as $WMS_{DINO\text{-}ViT\text{-}B/8}$ in the table) and also report the CLIP Score [27] for image-text alignment. In addition, we compute and report the mean similarity scores for the Medieval and Cubism reference images, denoted as $MS_{med}$ and $MS_{cub}$, respectively. These metrics quantify the alignment of generated images with each reference style individually, providing insight into how well each style is preserved in the blended output.

**Setup.** We generate images by blending two reference styles (e.g., *Medieval* and *Cubism*) using several weight configurations: $(0, 1)$, $(0.15, 0.85)$, $(0.25, 0.75)$, $(0.5, 0.5)$, $(0.75, 0.25)$, $(0.85, 0.15)$, and $(1, 0)$, and we fix the guidance scale to 10. In our experiments, the weight configuration, e.g. $(w_{med}, w_{cub})$, determines the relative influence of the Medieval and Cubism styles on the generated image. For example, $(0, 1)$ yields full Cubism and $(1, 0)$ full Medieval; $(0.5, 0.5)$ produces an even blend, while $(0.25, 0.75)$ or $(0.75, 0.25)$ skew the output toward Cubism or Medieval, respectively. Inspired by the StyleAligned[20] approach, we also created a dataset for the textual input prompts, composed of seven sets of nine images, using ChatGPT. This dataset comprises a diverse collection of stylistic descriptions and artistic references, enabling systematic evaluation of our blending methods across a wide range of style fusion scenarios.

**Results.** Table 3 summarizes our findings. The rows $\{0, 1\}$ and $\{1, 0\}$ represent the *StyleAligned* [20] baseline with no multi-style blending. From Tab. 3, SLI outperforms Linear interpolation in multi-style alignment ($WMS_{DINO\text{-}ViT\text{-}B/8}$) while maintaining better CLIP scores. The differences become more pronounced in blended scenarios (e.g., $\{0.5, 0.5\}$) where linear blending struggles to stay on the latent manifold, leading to suboptimal style fusion.

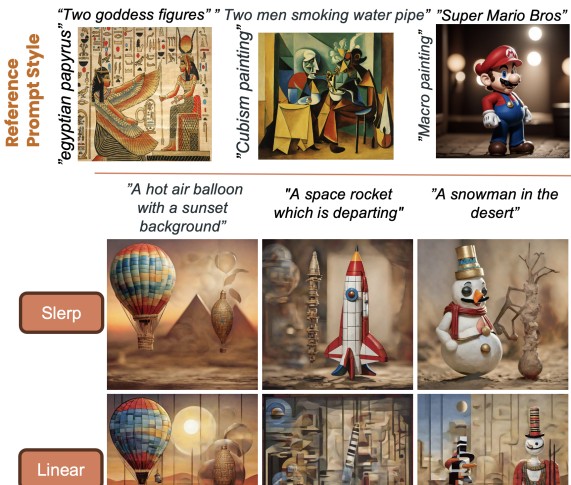

Figure 4. Linear vs. SLI Blending with three styles. Linear interpolation can fall off the meaningful latent manifold, introducing artifacts. On the other hand, SLERP provides always robust results.

## 4.2. Multi-Modal Content Fusion Module Inclusion

Although our primary focus is on multi-style blending, we also evaluate an optional *Multi-Modal Content Fusion* module (see Fig. 2) to demonstrate the advantages of enriching the input prompt beyond simple text, defined in our archi-

Table 4. Ablation Study: Attention Scaling vs. Non-Scaling (Z-SASLM's SLI $\{0.5, 0.5\}$ Blending)

| Scaling Parameters (*Cubism*) | $\text{MS}_{\text{med}}$ | $\text{MS}_{\text{cub}}$ | $\|\text{MS}_{\text{med}} - \text{MS}_{\text{cub}}\|$ | $\text{WMS}_{\text{DINO-ViT-B/8}}$ | $\text{CLIP}_{\text{score}}$ |
|---|---|---|---|---|---|
| $\{\log(1), 0.125\}$ | 0.36096 | 0.40155 | 0.04059 | 0.38126 | 0.31817 |
| $\{\log(1), 0.25\}$ | 0.35660 | 0.39909 | 0.04249 | 0.37784 | 0.28070 |
| $\{\log(1), 0.5\}$ | 0.36150 | 0.42156 | 0.06006 | **0.39153** | **0.31434** |
| $\{\log(1), 0.75\}$ | 0.36272 | 0.41224 | 0.04952 | 0.38748 | 0.31706 |
| $\{\log(2), 1\}$† | 0.33432 | 0.45480 | 0.12048 | 0.39456 | 0.30885 |

†*: Without Attention Scaling*

tecture as *"Single-Content Textual Prompt"*. In many creative tasks, a single-context prompt may lack the nuance necessary to capture the full spectrum of an intended artistic vision. By incorporating additional modalities—such as visual cues, audio transcripts, musical mood descriptors, and real-time weather data—we provide a richer and more informative conditioning signal that enhances style alignment and improves image quality.

In our approach, each modality is first converted into text: a photo is described via *BLIP[28]-based Image-to-Text*, spoken content is transcribed with *Whisper[29]*, music is interpreted via *CLAP[30]* combined *with cosine similarity* matching, and weather data is retrieved through the *OpenWeather API[31]*. This process has been employed to continue avoiding to perform the fine-tuning step. These diverse textual snippets are then concatenated and further condensed using *T5[32]-based paraphrasing* to produce a compact, yet semantically rich, "Multi-Content Textual Prompt" that is fed into the SDXL pipeline.

As shown in Fig. 5, our Z-SASLM Blending method maintains coherent style alignment even under these richer conditions, demonstrating the advantage of multi-modal fusion in guiding the generation process.

### 4.3. Scaling vs Non-Scaling

We next investigate the impact of attention scaling $(\mu, \sigma)$ on Z-SASLM's SLI blending. Using an equal weight configuration $\{0.5, 0.5\}$ for two reference styles, we vary the scaling parameters for the more prominent style (*Cubism*). Table 4 shows that without scaling, the difference between mean similarities $|\text{MS}_{\text{med}} - \text{MS}_{\text{cub}}|$ is significantly larger, implying style imbalance. Rescaling helps mitigate dominance by famous styles, ensuring a more balanced blend. CLIP scores remain fairly stable. You can also observe this behavior from Figure 6, noting the predominance of the Cubism style w.r.t. the medieval one during image generation.

### 4.4. Guidance Ablation

Finally, we vary the guidance scale (5 to 30) under SLI blending with weights $\{0.5, 0.5\}$. Table 5 indicates that higher guidance often increases the Weighted Multi-Style DINO VIT-B/8 metric but may also increase

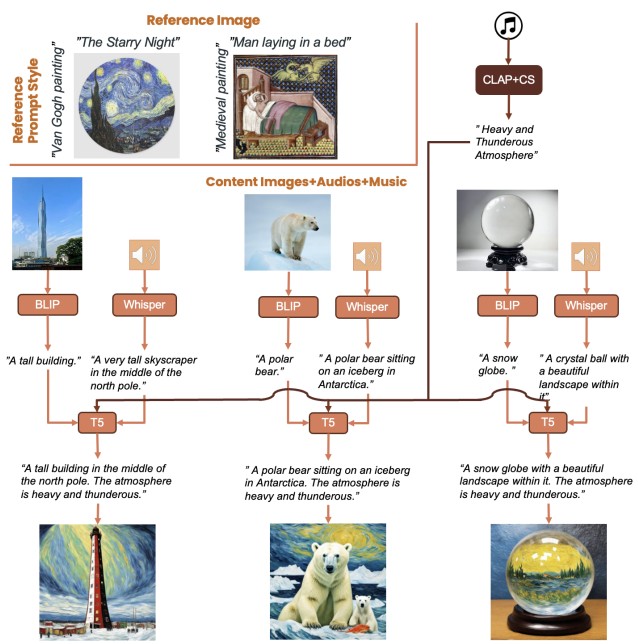

Figure 5. Z-SASLM with Multi-Modal Content Fusion. Our SLI blending (with $(w_{\text{med}}, w_{\text{vangogh}}) = (0.15, 0.85)$) preserves style alignment despite varied contextual cues (image, audio, music).

$|\text{MS}_{\text{med}} - \text{MS}_{\text{cub}}|$. This suggests a trade-off: stronger guidance can boost style fidelity but might amplify one style more than the other. Overall, moderate guidance (15–20) strikes a good balance. This behavior can also be observed directly through Figure 7, where you can observe how increasing the guidance scale progressively amplifies the stylistic influence from the two reference images. At low guidance (5), each generated image remains relatively faithful to its textual prompt. As guidance rises (10–15), characteristic shapes, colors, and patterns drawn from the reference images become more evident. By higher guidance levels (20–25), the stylization is more aggressive; the original content is still recognizable, but geometric forms, bold outlines, and subdued palettes—drawn from the references—dominate the overall look. In other words, strong guidance enhances style fidelity but can overshadow fine details of the prompt, creating images that lean more heavily toward the shared artistic signature of the references.

| Z-SASLM {0.5, 0.5} | | | | | |
|---|---|---|---|---|---|
| Guidance | $MS_{med}$ | $MS_{cub}$ | $|MS_{med} - MS_{cub}|$ | $WMS_{DINO\text{-}ViT\text{-}B/8}$ | $CLIP_{score}$ |
| 5 | 0.37721 | 0.38623 | 0.00902 | 0.38172 | 0.31420 |
| 10 | 0.36149 | 0.42156 | 0.06007 | 0.39153 | 0.31434 |
| 15 | 0.35610 | 0.43315 | 0.07705 | 0.39463 | 0.31681 |
| 20 | 0.38218 | 0.45844 | 0.07626 | **0.42031** | **0.31656** |
| 25 | 0.36966 | 0.44971 | 0.08005 | 0.40968 | 0.31554 |
| 30 | 0.36350 | 0.46818 | 0.10468 | 0.41584 | 0.31316 |

Table 5. Guidance Ablation Study for Z-SASLM $\{0.5, 0.5\}$ Blending.

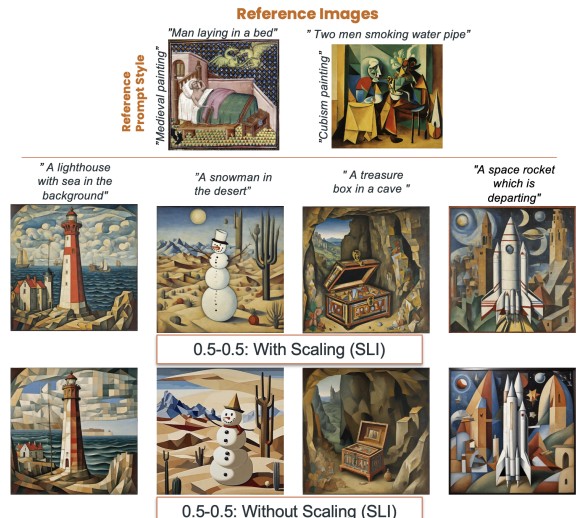

Figure 6. Attention score rescaling effect under SLI blending.

## 5. Conclusions

We have presented the novel Z-SASLM framework for multi-style blending in diffusion models, focusing on a Spherical Linear Interpolation (SLI) approach that respects the non-Euclidean geometry of the latent space. Our experiments show that Z-SASLM outperforms the Linear baseline (inspired by StyleGAN2-ADA) across various style weight configurations and under multi-modal content prompts. Additionally, we introduced a Weighted Multi-Style DINO ViT-B/8 metric to quantify style consistency, demonstrating the superiority of SLI in navigating complex latent manifolds. Although our main emphasis is on multi-style blending, we also illustrated that Z-SASLM's solution remains robust when additional modalities (audio, music, weather) are fused into the prompt. Future work will explore more efficient attention mechanisms and extended style references for large-scale applications.

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
