# OpenReview forum: "Z-SASLM: Zero-Shot Style-Aligned SLI Blending Latent Manipulation"
_thecvf.com/CVPR/2025/Workshop/CVEU — CVPR 2025_

### Official Review · Reviewer_EN4D · 2025-03-16
**review of 17**

**Rating:** 2
**Confidence:** 4

**Review:**

This paper proposes a Z-SASLM, a zero-shot style aligned slerp Blending Latent manipulation pipelines for style blending. Based on the StyleAligned pipeline, this method applies Spherical Linear Interpolation to blend features from multiple style images, and design a rescaling method to avoid over-stylization. It also proposes a weighted multi-style DINO VIT-B/8 metric to measure the similarity between multiple styles. The results look fair, and the main idea is easy to follow.

However, the **novelty** is limited to me. To use slerp is not new in diffusion. For example, [1] uses slerp to interpolate latent noise in diffusion models. In addition, as for the rescaling method, it is unclear how and why the parameters are set in Eq. (14).

In **experiment**, there are two styles of Medieval and Cubism are reported. Why using these two styles and I believe more styles should be evaluated to make the conclusion more convincing. The visual comparison results with Linear are limited. Only three images on two styles.

Other issues:
1. The claimed contributions 2 and 3 are less discussed in the introduction.
2. Line 98, the authors claim that `These approaches (GANs) suffer from several limitations: they assume a flat (Euclidean) latent space that fails to capture the curved geometry of high-dimensional representations`. However, GAN, e.g., StyleGAN is very linear in latent space, making it easy to manipulate faces. The claim is not true.
3. The order of Fig. 3 and Fig. 4 should be exchanged.
4. Why scale is set to 10 in Line 351 while the optimal scores are obtained when scale is 2o in Table 5?
5. The impact of audio, music and weather seems insignificant from the results.

> [1] Diffmorpher: Unleashing the capability of diffusion models for image morphing. CVPR 24

---

### Official Review · Reviewer_Tzd1 · 2025-03-20
**A new multi-style blending for style conditioning**

**Rating:** 4
**Confidence:** 4

**Review:**

**Strengths**

++ The paper proposes a new spherical linear interpolation (SLI) blending that can combine multiple reference-style images in a weighted manner and preserve style fidelity.

++ Paper is well written and the multi-modal content fusion evaluation is extensive.

**Weakness**

-- More qualitative comparisons against previous methods should be provided.

---

### Decision · Program_Chairs · 2025-03-25

**Decision:**

Accept

**Comment:**

The paper proposes Z-SASLM, a zero-shot style blending approach using spherical linear interpolation (SLI) for multi-style conditioning. Reviewers appreciated the clear writing, intuitive method, and extensive evaluations. However, concerns included limited novelty, insufficient qualitative comparisons, and unclear parameter choices.

Despite these weaknesses, the overall strengths in addressing a practical task effectively justify acceptance. The paper is thus accepted. Authors should provide clearer justifications for parameter settings and expanded qualitative comparisons in the camera-ready version.